# Model Issues Regarding Modification of Fuel Injector Components to Improve the Injection Parameters of a Modern Compression Ignition Engine Powered by Biofuel

Jacek Eliasz, Tomasz Osipowicz *, Karol Franciszek Abramek and Łukasz Mozga

Department of Automotive Engineering, West Pomeranian University of technology, 70–310 Szczecin, Poland; Jacek.Eliasz@zut.edu.pl (J.E.); Karol.Abramek@zut.edu.pl (K.F.A.); ml23722@zut.edu.pl (Ł.M.)

* Correspondence: tosipowicz@zut.edu.pl; Tel.: +48-503420350

**Abstract:** This article presents a theoretical analysis of the use of spiral-elliptical ducts in the atomizer of a modern fuel injector. The parameters of the injected fuel stream can be divided into quantitative and qualitative. The quantitative parameter is the injection dose amount, and the qualitative parameter is characterized by the stream of injected fuel (width, atomization, opening angle, and range). The purpose of atomizer modification is to cause additional flow turbulence, which may affect the stream parameters and improve the combustion process of the combustible mixture in a diesel engine. The spiral-elliptical ducts discussed here could be used in engines powered by vegetable fuels. The stream of such fuels has worse quality parameters than conventional fuels, due to their higher viscosity and density. The proposal to use spiral-elliptical ducts is an innovative idea for diesel engines.

**Keywords:** fuel injector; CI engine; biofuel; fuel combustion

## 1. Introduction

The research problems of modern compression ignition engines concern issues related to ecology. The most important aspect is the emission of pollutants in exhaust gases. Experiments on reducing the toxicity of exhaust gases have been carried out in various directions, such as the use of electronic engine management systems, the use of exhaust after-treatment systems, various modifications of engine components, and the use of alternative fuels. In order to meet stringent standards regarding the emission of pollutants in the exhaust gases of diesel engines, a common rail fuel supply system has been introduced. The advantage of this system is the ability to adjust the injection torque and pressure of the engine operating conditions by separating the fuel accumulating element (injection pump) with fuel injectors with a pressure accumulator (rail). Various design solutions for the Common Rail system have been developed, but their operating principle is the same. The essence of this solution is to divide the injection doses into one engine cycle, and to control the desired pressure and fuel injection moment. The purpose of this solution to appropriately organize the combustion process of the combustible mixture in the engine compartment so that the emission of toxic substances to the atmosphere, such as nitrogen oxides, carbon monoxide, unburned hydrocarbons and solid particles, is reduced. The element of the injection system responsible for organizing the combustion process of the combustible mixture in the cylinder workspace is the injectors. Their task is to distribute and atomize fuel in the combustion chamber. Two aspects of the injection process are considered: Quantitative and qualitative. The quality of the injected fuel stream plays an important role in the process of creating a combustible mixture. Its physical properties, such as viscosity and density, are responsible for this parameter. Fuels of plant origin have a higher viscosity and density, which affects the parameters of the stream, such as

atomization, width, length, or angle of opening. However, it is possible to change these parameters by using spiral-elliptical ducts on the non-working part of the sprayer. This work analyzes the possibility of using this solution in the injection systems of compression ignition engines.

Improving the quality parameters of the fuel stream can be achieved by increasing the pressure in the injection system. The issues related to the influence of pressure were analyzed in Wang et al. work [1]. They showed that the range of the stream depends on the pressure in the system. The most favorable penetration result was obtained from a pressure of 150 MPa to 210 MPa. The spray angle is the largest at low pressures. With increasing pressure, it decreases, and remains constant. The speed of fuel outflow from the atomizer can reach 320 m/s at a pressure of 250 MPa. Authors Han et al. [2] characterized how fuel properties affect the droplet size during spraying. Conventional diesel oil, rapeseed oil methyl ester, rapeseed oil ethyl ester, and biofuel made from coconut oil were used for their research. Studies have shown that fuel with higher surface tension and viscosity, such as ethyl and methyl ester of rapeseed oil, is characterized by a greater ability to penetrate the stream, but the width of the opening angle and the flow area of the fuel spray is lower. Coconut oil methyl ester and conventional diesel oil were characterized by better atomization and spray surface. Microscopic studies have also shown that fuels with lower viscosity and surface tension (diesel oil coconut methyl ester) have smaller Sauter diameters (SMD) in the sprayed stream. Research conducted by Anis, et al. [3] showed that preheating the fuel improves spraying parameters. The analysis was carried out for pure biofuels and mixtures of biofuels with conventional fuel. The test results showed that heating up the fuel in fuel injectors to up to 343 K improves the quality characteristics of the injected fuel stream, and heating up to 323 K in the injection pump increases its efficiency. Other research Han et al. [2] compared how the internal velocity of the flow influences the division and fragmentation of fuel droplets during atomization. During fuel injection into the engine compartment, turbulent flows occur, which affect the quality parameters of the stream. The purpose of the simulations was to help understand the process of fragmentation of fuel droplets in the combustion chambers of a compression ignition engine. The analysis showed that intensity of turbulent flow affects the process of fragmentation of the fuel stream. Increasing the pressure in the atomizer increases the fuel speed at the atomizer outlet, which increases turbulence. In another article, Anantha et al. [4], the process of injection of two diesel fuels and soybean oil methyl ester was compared. Computer analysis of fuel flow through the atomizer and simulation of the atomization process was carried out for a Bosch 1.0 fuel injector at 135 MPa injection pressure, 1 MPa overflow pressure, 300 K ambient temperature, and 0.5 ms injection time. Two types of fuel atomizers, with cylindrical and conical holes, were used for the tests. The analysis showed that a much higher occurrence of cavitation appears when using cylindrical holes in fuel atomizers, regardless of the type of fuel. The fuel flow through the atomizer is higher for tapered holes, though minimally so for diesel, due to the lower density compared to soybean oil methyl ester. The average fuel flow rate is higher for diesel and comparable for the type of holes in the fuel atomizer. This is due to the lower viscosity of conventional fuel. Studies have shown that plant biofuels have lower tendencies towards cavitation, due to their higher density and viscosity Bishop et al. [5] and Yu et al. [6]. Observation of the stream of injected fuel showed that the angle of opening with the use of cylindrical holes in the atomizer is similar for diesel and vegetable fuel, while with the cone atomizer it is larger for biodiesel, because it produces a larger cloud during scattering in the combustion chamber. The largest range of the fuel stream was recorded for diesel with tapered holes, while for biofuel the design of the hole was irrelevant. The Sauter diameter was definitely lower for conventional fuel regardless of the type of hole used in the fuel atomizer. The difference during fuel flow through the atomizer openings and the atomization process is mainly influenced by the density and viscosity of the tested fuels Bishop et al. [5]. Research carried out in Tinprabath et al. [7] presents a theoretical and experimental analysis of flow in the atomizer, and the injection characteristics of conventional fuel without bioadditives (B0) and biofuel (B100). The simulation results showed that the phenomenon of cavitation and turbulence during flow is more common with conventional fuels. As the fuel pressure in the atomizer increases, the kinetic energy of the liquid increases, which increases turbulence and cavitation.

Authors Tinprabath et al. [7] present an experimental analysis of the impact of diesel oil, vegetable fuel, and their common mixtures on the injection flow rate. Bosch piezoelectric injectors and five types of fuel were used for the tests: Diesel oil, winter diesel oil, mixtures of diesel oil and B20 biofuels (80% diesel fuel, 20% biofuels), B50 (50% diesel fuel, 50% biofuels), and clean B100 biofuel. The fuel injector pressure was 30–60 MPa at room temperature, at which no evaporation of the mentioned fuels takes place, and below zero ($-5$ °C and $-8$ °C). The experiment showed that low ambient temperatures do not affect the auto-ignition delay period. Fuel viscosity affects the duration of the injection, therefore, there are differences in the injection dose for individual fuels. At positive temperatures for B100 vegetable fuels, the flow rate decreases linearly until the viscosity increases. At negative temperatures for fuel with additives (diesel fuel and diesel fuel winter mixed with biofuel), viscosity, measured at atmospheric pressure, is not the only property that affects the flow rate. Pure diesel, B20 and B50 had similar viscosities. Only winter diesel fuel had better parameters.

The injection parameters are affected by the physico-chemical properties of the fuel, such as density, viscosity, surface tension, the heat of evaporation, boiling point, liquidity point, and cetane number Pandey et al. [8]. Density is a parameter directly proportional to the volume elasticity coefficient. Factors, such as density, speed of the propagating wave sound, and the volumetric elasticity coefficient affect the moment of fuel injection into the combustion chamber (injection advance angle). The higher the fuel density and volumetric coefficient of elasticity, the earlier the fuel begins to oxidize in the cylinder space, which causes the cylinder to have a higher temperature and higher nitrogen oxide emissions in exhaust gases Pandey et al. [8]. Research Knothe [9] has shown that biodiesel made of saturated fatty acids (low iodine number) has a higher cetane number, but worse low temperature properties, and biodiesel made of unsaturated fatty acids, inversely has a low cetane number and better properties at low temperatures. In addition, the volume modulus has a higher value for unsaturated fatty acid methyl esters than for saturated fatty acid methyl esters. Saturated fatty acid esters have a lower density, due to the smaller number of carbon bonds in the chain Pandey et al. [8]. Kinematic viscosity determines the liquid's fluidity. The fluidity decreases as the kinematic viscosity increases. This is another very important parameter affecting the quality parameters of the fuel injection process into the combustion chamber. This property affects the atomization of the fuel stream. The higher the kinematic viscosity, the lower the spray angle and the droplet diameter during the injection process. This results in the deposition of soot in the combustion chamber of the engine and its components parts, contamination of the engine oil, and startup problems at lower temperatures. In work Knothe [9], the influence of the number of fatty acid bonds on the magnitude of kinematic viscosity was examined. Studies have shown that the chain length, position and number of double bonds affect kinematic viscosity and ease of oxidation. The longer the chain, the more the viscosity increases, the greater the number of unsaturated fatty acids in methyl esters of vegetable oils, and the smaller the decrease. Too high a viscosity of the fuel causes earlier ignition, which adversely affects the combustion process and causes increased emissions of nitrogen oxides into the atmosphere Pandey et al. [8]. The boiling point affects the evaporation properties of the fuel. The smaller it is, the easier it is to create a combustible mixture. The fuel distillation temperature increases with the content of carbon molecules in the fuel, and decreases with the increase of unsaturated fatty acids Kegl et al. [10], Knothe [9]. The liquidity point determines how low the fuel can flow. The disadvantage of vegetable fuels is insufficient liquid properties at low temperatures. This is affected by the chain length of higher fatty acid methyl esters. If the ester contains more saturated fatty acids, the liquidity temperature increases, and if it contains more unsaturated fatty acids, it decreases Pandey et al. [8]. The parameters determine the injection properties of a modern fuel injector, such as injection time, injection advance angle, injection dose, fuel injection delay, and atomizer opening pressure, and play an important role in the combustion process. The fuel injection time depends on the chemical properties of the fuel, especially its calorific value. Lower calorific fuels require longer fuel injection times, while higher ones require the opposite. Longer injection times cause an incomplete combustion process, which increases the emission of carbon oxides, soot, and hydrocarbons in the exhaust gas. Another parameter determining the fuel

injection properties is the injection advance angle. It is affected by the physical fuel properties, such as the density, kinematic viscosity, and volumetric coefficient of elasticity of the fuel. If the mentioned parameters increase, then the attenuation of fuel in the injector decreases, and it causes faster needle rising in the atomizer, which extends the injection advance angle Salvador et al. [11]. This results in increased pressure in the combustion chamber and a higher emission of nitrogen oxides. Density, viscosity and sound velocity value have an impact on the compressibility factor of the fuel during flow through the fuel atomizer holes, but they do not affect the Reynolds number Salvador et al. [12]. The fuel injection CFD simulation was carried out in two phases: Incompressible flow at constant liquid properties and compressible flow at liquid properties, calculated locally as a function of flow pressure conditions. Analyzing the research carried out by the authors, it can be stated that the issues related to fuel flow through the injector atomizer should be considered in terms of compressibility of the liquid, depending on the amount of pressure and flow temperature. At low system pressures, the flow is moderate. The liquid is in a state of transition between the laminar and turbulent flow Payri at el. [13]. The injection dose factor is then dependent on the Reynolds number. However, when the pressure in the system begins to increase, the flow passes into the turbulent phase, then the injection dose factor is independent of the Reynolds number Salvador et al. [14]. In work Desantes et al. [15], the process of injection of methyl esters of rapeseed oil mixed with diesel oil was compared in the proportions—5% bio-additive (B5), 30% bio-additive (B30) and pure methyl ester (B100). Studies have shown that the parameters of the fuel stream for B5 and B30 fuels are similar. The pure rapeseed oil methyl ester stream has a higher flow rate, due to the higher density, but similar momentum. The analysis of the fuel injector operation showed that its speed depends on the fuel used. The increased viscosity of the biofuel causes the needle to rise more slowly. This is very important when generating small doses of fuel, which depend mainly on the dynamics of the fuel injector. It was noticed that clean biofuel has a slightly larger penetration range, but a narrower angle of opening of the stream, which affects the dose of injected fuel Saltas et al. [16]. Due to the increased density, the speed of the injected fuel is lower, and the surface tension causes poorer atomization of the droplets. These factors cause worse mixing conditions with air and directly influence the combustion process. Biofuels have an inclination towards dampness absorption Sundus et al. [17] and Othmana et al. [18]. Research conducted by Antonov et al. [19] discussed the spray effects two fluids: Diesel and water, and rape oil and water. The results showed that the micro-explosion of drops in the biofuel stream needs more energy because of higher viscosity and surface tension compared to diesel fuel. The structure of primary and secondary breakup droplets depends on the Weber number Minakov et al. [20].

After analyzing the literature on the subject of this work, it can be stated that phenomena occurring during the course of fuel atomization, such as the formation of waves, their amplitude increase, and loss of stability, are affected by fluid vibrations during motion. The frequency of vibrations varies, and their causes can be external and internal. Internal factors are disturbances arising in the atomizer itself, such as liquid turbulences, atomizer vibrations (because the work of the needle), expansion of the liquid, due to pressure changes, and disturbances in the movement of liquids on the unevenness and edges of holes. External factors depend on the liquid speed, gas density, liquid surface and pressure. On the basis of the authors' preliminary research Osipowicz et al. [21], it can be concluded that the implementation of changes to the non-working part of the needle affects the course of the process of atomizing fuel in the combustion chamber. Fuels of plant origin, due to their physical properties, have poorer spray ability in comparison with conventional diesel oil. Spiral-elliptical ducts may cause additional turbulence during the liquid flow through the atomizer immediately before the injection process. Due to the additional turbulence caused, particles may break down, and the quality of the fuel stream may improve, which will affect the combustion of the combustible mixture. Increased flow turbulence can locally increase the temperature of the liquid in the atomizer. By analyzing the literature, we can conclude that temperature has a significant impact on the density and viscosity of biofuels. The proposed changes are innovative and have not been used anywhere before.

None of the above works analyzed how the additional turbulence caused by the fuel flow before flowing into the atomizer holes will affect the atomization, range, angle, and speed of the injected fuel stream. Additionally, it is possible to change parameters directly in the fuel atomizer, due to turbulence and increased temperature. In previous studies in this field, no attempts have been made to carry out modifications of fuel atomizers. Increasing the pressure in the system, heating the fuel, and changing the design of the combustion chambers are known strategies that have been extensively explored. Therefore, the innovative element of this project is the analysis of the fuel flow in a modified atomizer.

The phenomena of fuel flow in the atomizer are variable. During the injector operation, the atomizer closes and opens continuously. The type of flow depends on the pressure in the whole system. In modern Common Rail systems, the pressure in the fuel tank oscillates between 25–200 MPa, and the value depends on the engine load.

This paper analyzes an electromagnetic fuel injector of the Bosch 1.0 generation, working on pressures in the range of 30–135 MPa. The fuel flow model was created based on a test of the injector on a Zapp Carbontech CRU2i test bench.

## 2. Impact of Fuel Injection Parameters on the Combustion Process in a Compression Ignition Engine

The parameters of the injected fuel stream have a significant impact on the combustion process in a compression ignition engine. The process of creating a combustible mixture consists of atomizing and spreading the fuel stream, heating, evaporation, and mixing of the fuel with air. The preparation of the combustible mixture lasts from the moment of starting the fuel injection to the end of its combustion. Combustion of a combustible mixture in a compression ignition engine depends on the quantitative and qualitative characteristics of the injection, design of the combustion chamber, fuel properties, speed and direction of the load propagation, location of fuel injectors, and mutual orientation of the fuel streams. Ignition of atomized fuel is a chain; a multi-stage process. The first ignition foci are formed according to the volumetric combustion process. The flame spreads, and the combustion of the fuel-air mixture prepared during the auto-ignition delay period begins. The combustible mixture goes into the diffusion combustion stage and afterburning takes place. The process of fuel combustion in a diesel engine takes place in several stages.

The first stage of the combustion process is the auto-ignition delay period ($\tau_s$). It is the time from the beginning of fuel injection until the appearance of the first auto-ignition foci (on the indicator graph, the point at which the pressure in the cylinder, due to heat release is greater than the pressure in the cylinder when compressing air without injection). The auto-ignition delay is the time when the stream breaks down into droplets, partial evaporation and mixing of fuel vapors with air, and a period of acceleration of chemical reactions. This stage is intended to be shortened, because a shorter delay of self-ignition causes less fuel flow to the chamber, which causes a slower pressure increase in the combustion chamber. A pressure increase during the combustion process that is too rapid increases the combustion temperature, which promotes increased emission of nitrogen oxides in exhaust gases, increases the impact loads of engine components, and increases the noise associated with the combustion process. The average rate of pressure rise should not exceed 0.3–0.8 MPa/°OWK. The following factors affect the auto-ignition delay period.

1.  Cetane number of the fuel, which determines the ability of a given fuel to ignite. The higher its value, the better the fuel's flammable properties. The cetane number of fuels can be increased by using various additives.
2.  The type of combustion chamber affects the self-ignition delay because there are differences in the distribution of fuel in the space and layer at the wall, and in the temperature distribution of the walls of the combustion chamber.
3.  The fuel pressure and temperature at the beginning of the fuel injection process affect the first stage of the combustion process. Increasing the pressure, and as a result, the temperature of the lower injection timing shortens the auto-ignition delay.

4. Quantitative and qualitative characteristics of the fuel injection. The dose and shape of the stream affect the auto-ignition delay period. Correct injection rates (pilot, basic) generated by fuel injectors and appropriate atomization of the fuel jets shorten the auto-ignition delay.

5. Increasing the engine speed improves atomization and increases the pressure and temperature in the combustion chamber, which reduces the ignition delay.

The targeted intensity of the charge movement in the combustion chamber shortens the auto-ignition delay period.

The auto-ignition delay period according to Heywood is determined by the following relationship [9],

$$\tau_s = (0.36 + 0.22 \cdot c_m) \cdot \exp\left[E_a\left(\frac{1}{RT_2} - \frac{1}{17.19}\right) \cdot \left(\frac{21.2}{P_2 - 12.4}\right)^{0.63}\right],$$ (1)

where $P_2$ is the temperature in the combustion chamber, $T_2$ is the pressure in the combustion chamber, $E_a$ is the amount of activation energy, and $c_m$ is the average piston speed.

The activation energy value is calculated from the formula

$$E_a = \frac{618.840}{LC + 25} \ [\text{J/mol}],$$ (2)

where LC is the cetane number.

Based on relationship (1), it can be stated that increasing the cetane number of fuel extends the delay of auto-ignition. According to theoretical considerations, the quality of fuel atomization in the combustion chamber does not affect the auto-ignition delay period. However, the considerations apply to older design engines operating at low pressures in the injection system, and the old generation fuel injectors. The Common Rail system works at working pressures of 25–200 MPa. Phenomena inside the system are very variable. The occurring turbulent flow of liquid in the fuel atomizer means that the atomization, width, range, and angle of the stream of flowing fuel have different values than in classic systems. Additionally, the changes made to the fuel atomizer can affect the additional turbulence of the liquid and improve its mixing with air. The second stage of the combustion process, called kinetic combustion (rapid combustion), lasts from the moment of the ignition of the combustible mixture until reaching the maximum pressure in the cylinder. During this phase, oxidation of part of the fuel–air mixture prepared during auto-ignition delay occurs, resulting in rapid heat release and pressure build-up. At the end of this stage, the combustion process is limited by the speed of mixing of the fuel with air. The following factors influence the kinetic combustion period.

1. The injection dose introduced during the auto-ignition delay period and during kinetic combustion, and the characteristics of the injected fuel stream. A smaller amount of fuel supplied during the auto-ignition delay causes a smaller value of pressure increase in the combustion chamber in relation to the crankshaft rotation. If the atomization is finer, the faster the first doses of fuel mix with the air, and the more gently the pressure in the cylinder in the second phase increases.

2. The design of the combustion chamber significantly affects the development of the second stage, and in particular, its impact on the auto-ignition delay period and the amount of air-fuel mixture prepared for combustion after it has started.

3. Increasing the charge movement speed to a certain value in the combustion chamber intensifies heat release in the kinetic combustion phase.

4. When the engine speed increases, the atomization improves, the auto-ignition delay decreases, the charge movement speed in the combustion chamber increases, and pressure and temperature increase. These factors accelerate chemical reactions.

5. By reducing the engine load, the duration of the second stage is shortened, resulting in a reduction of the fuel dose and a shorter time for its supply to the cylinder.

The third stage of combustion begins when the reaction speed is much higher than the mixing speed of the reactants, then the duration of the combustion process depends on the mixing speed of the reactants. This period is called diffusion combustion. It begins at the moment of the greatest pressure in the cylinder, and continues until the maximum temperature is reached. In modern diesel engines, the temperature reaches a maximum of 20–40 °C according to GMP. This is due to the fact that intense heat production is caused by the second period of the combustion process. The third stage of combustion of the combustible mixture in a compression ignition engine depends on the following factors.

1. The use of engine boost increases the amount of heat generated. By increasing the engine boost level, the duration of the third phase is extended. This is related to the amount and speed of heat released.
2. Injection dose size and quality of the fuel atomization process after starting the combustion process. At low engine loads, the fuel injection ends before the beginning of the third combustion phase, then the amount of heat released is small.
3. By increasing the speed of movement of the load in the cylinder to a certain value, heat release in the third stage increases. However, too much air turbulence reduces heat generation, due to deterioration of fuel decomposition in the combustion chamber, and movement of combustion products from the zone of one fuel stream to another. This may result in incomplete combustion and an increase in smoke opacity.
4. Increasing the engine speed increases the supply, fuel atomization, and load speed in the combustion chamber, which shortens the third stage of combustion.

The fourth stage of combustion in a compression ignition engine called afterburning begins when the maximum temperature of the cycle is reached, and lasts until the end of heat release. A soot burning reaction occurs during this stage. The following factors affect the course of the fourth stage.

1. The phenomenon of fuel falling on cool surfaces of the cylinder space, which causes the extension of the duration of this stage.
2. Supercharging the engine leads to a prolongation of the period of combustion of fuel, due to the longer fuel injection time and the deterioration of the spread of the fuel in the combustion chamber.
3. The spraying process in the final stage of combustion. The large maximum diameter of the fuel droplets results in extended fuel injection, which means that the afterburning time of the combustible mixture is longer. This deteriorates the use of thermal energy resulting from combustion, which affects the operation and reliability of the engine by depositing carbon combustion deposits and coking the fuel atomizers.
4. Turbulent charge fluctuation contributes to improving the mixing of fuel and air.

Analyzing the combustion process of the fuel–air mixture in a diesel engine, it can be concluded that the qualitative characteristics of fuel atomization in the cylinder space affect each stage of combustion.

## 3. Presentation of the Modifications Made to the Fuel Injector Sprays

The purpose of spray modification is to improve the fuel injection process in a modern diesel engine with a Common Rail system powered by vegetable fuel. The fuel injection process can be considered in terms of quantity and quality. The quantity, describing the quantitative aspect, is the injection dose, while the qualitative aspect includes the shape, range, width, opening angle, speed, and atomization of the stream of fuel flowing from the atomizer. Figure 1 shows a modified fuel atomizer with spiral-elliptical ducts Osipowicz [22].

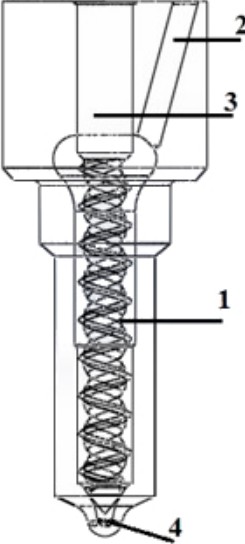

**Figure 1.** The modified fuel atomizer. (**1**) Channels in the non-working part of the needle, (**2**) the fuel supply channel, (**3**) the precision atomizer pair, (**4**) the injection holes [22].

The element of the fuel injector responsible for the correct process of fuel injection into the combustion chamber is the atomizer (Figure 1). It consists of a body and a needle. The fuel flows into the needle chamber through channel 2, then flows around the non-working part of needle 1. Annular channels made on the non-working part of the needle, as a result of the reciprocating movement of the needle, cause additional fuel turbulence. When the cone valve 4 opens the holes in the fuel atomizer, injection into the combustion chamber takes place. The purpose of the atomizer modification is to improve the atomization process of vegetable fuels in the combustion chamber of the engine, due to the turbulence caused by the atomizer and the accompanying phenomena. The parameters affecting the quality of the injected fuel stream are viscosity and density. Table 1 presents selected physical parameters of conventional and vegetable fuels. As can be seen, the viscosity and density of vegetable fuels have higher values than conventional fuel, therefore, according to the authors, the use of spiral-elliptical ducts on the non-working part of the needle will affect the physical parameters of the fuel and improve the atomization process in the combustion chamber of the compression ignition engine.

**Table 1.** Physical parameters of diesel fuel B100 and raps oil

| Parameter | Diesel Fuel | Rapeseed Oil Methyl Ester | Rapeseed Oil |
|---|---|---|---|
| Density [kg/dm$^3$] 20 °C | 0.817–0.856 | 0.86–0.9 | 0.91–0.92 |
| Kinetic viscosity [mm$^2$/s] 15 °C | 2.90–5.50 | 6–9 | 68–97.7 |
| Ignition temperature [°C] | 20–84 | 111–175 | 317–324 |

One of the features of turbulent flows is the fluctuation of momentum and kinetic energy of turbulence Hinze [23], and the phenomenon of heat exchange during flow Elsner [24], Jiao [25]. The viscosity and density of liquids depending on the temperature. If the temperature increases locally under the influence of additional vortices, then the physical parameters of the fuel will also change, which will affect the spraying process Suh et al. [26]. Previous work related to the analysis of fuel flows in atomizers, and its atomization have not taken into account the induction of additional turbulence in the area of injectors. The concept of creating additional turbulence is innovative, and has not been used anywhere until now.

### 4. Fuel Flow Analysis in the Fuel Atomizer

In order to illustrate the phenomena occurring before the injection process, a fuel flow model was made for a standard and modified atomizer according to the received patent Osipowicz [19]. The analysis was performed using Solidworks Flow Simulation. This program enables the study of a wide range of fluid flow and heat transfer phenomena. The model has a demonstrative function, and its task is to examine whether the changes made have affected the fuel turbulence in the atomizer. The flow was tested when the atomizer was opened at the moment when the needle was raised. This experiment is the first stage of the design and implementation of spiral-elliptical ducts in modern fuel injectors.

Based on the tests carried out, it can be concluded that the tested fuel injectors are in working order. The simulation of the fuel flow through the atomizer was carried out for three measurements with the same actuation times, with 500 μs fuel injectors at 30, 60, and 120 MPa pressure (Tables 2 and 3).

**Table 2.** Measurements of the injection doses according to the standard test of the standard fuel injector.

| No. | Pressure [MPa] | Injection Time [μs] | Range [mm³/H] | Dosage [mm³/H] |
|-----|----------------|---------------------|---------------|----------------|
| 1 | 135 | 780 | 34.71–49.69 | 35.3 |
| 2 | 30 | 420 | 0.31–3.89 | 0.7 |
| 3 | 80 | 260 | 0.31–4.09 | 1.2 |

**Table 3.** Measurements of the injection doses according to the standard test of the modified fuel injector.

| No. | Pressure [MPa] | Injection Time [μs] | Range [mm³/H] | Dosage [mm³/H] |
|-----|----------------|---------------------|---------------|----------------|
| 1 | 135 | 780 | 34.71–49.69 | 34.8 |
| 2 | 30 | 420 | 0.31–3.89 | 0.9 |
| 3 | 80 | 260 | 0.31–4.09 | 1.1 |

During the tests the phenomena prevailing in the atomizer were described. The following figure shows the tip of the fuel injector divided into zones (Figure 2).

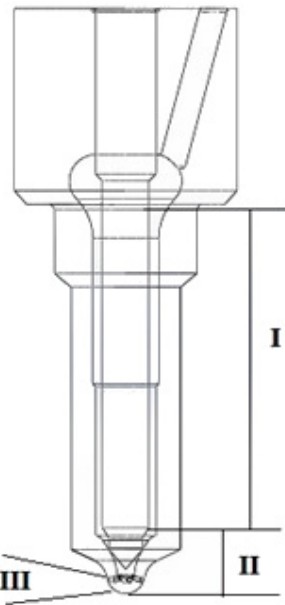

**Figure 2.** Fuel atomizer divided into zones: (**I**) The non-working part of the needle, (**II**) the needle well, (**III**) the outlet from the atomizer.

For the analysis of phenomena occurring in the atomizer of a modern fuel injector, general equations of the motion of viscous fluids, resulting from the momentum conservation principle, can be used. These phenomena are described by the Navier-Stokes equations in scalar form, where $V_{x, y, z}$ are fluid velocities in a Cartesian system, $\vec{V}$ is the fluid flow velocity, p is the pressure, X, Y, Z are the individual mass forces in the Cartesian system components, $\vec{F}$ is the individual mass force, and ρ is the fluid density:

$$\frac{dV_x}{dt} = X - \frac{1}{\rho} \cdot \frac{\partial p}{\partial x} + v \cdot \Delta V_x + \frac{1}{3} \cdot v \cdot \frac{\partial}{\partial x}(div\vec{V}), \tag{3}$$

$$\frac{dV_y}{dt} = Y - \frac{1}{\rho} \cdot \frac{\partial p}{\partial y} + v \cdot \Delta V_y + \frac{1}{3} \cdot v \cdot \frac{\partial}{\partial y}(div\vec{V}), \tag{4}$$

$$\frac{dV_z}{dt} = Z - \frac{1}{\rho} \cdot \frac{\partial p}{\partial z} + v \cdot \Delta V_z + \frac{1}{3} \cdot v \cdot \frac{\partial}{\partial z}(div\vec{V}), \tag{5}$$

and in vector form:

$$\frac{d\vec{V}}{dt} = \vec{F} - \frac{1}{\rho} \cdot grad(p) + v \cdot \Delta\vec{V} + \frac{1}{3} \cdot \gamma \cdot grad(div\vec{V}). \tag{6}$$

The following data was used to make the model: Channel (atomizer) parameters, flow rate (fuel injection dose), and fuel parameters (density and viscosity). The Bernoulli equation was used to calculate the speed of fuel flow through the atomizer and stream, where V, $V_1$, $V_2$ are fluid flows, $p_1$ and $p_2$ are the pressure, ρ is the fluid density, μ is the fluid dynamic viscosity coefficient, and $d_0$ is the flow dimension.

$$\frac{\rho V_1^2}{2} + p_1 = \frac{\rho V_2^2}{2} + p_2, \tag{7}$$

$$R_e = \frac{\rho \cdot V \cdot d_0}{\mu}. \tag{8}$$

Analyzing the figure, the pressure in the atomizer in zones I and II is at the same level. Based on relationships (7) and (8), the Reynolds number and fuel flow rate will also be the same. The flow parameters of the tested fuel injectors during the analysis are presented in the following table (Table 4).

**Table 4.** Fuel flow parameters through the atomizer during analysis.

| P [MPa] | Q [m³/s] | Re₁ | Re₂ | V₁ [m/s] | V₂ [m/s] |
|---------|----------|-----|-----|----------|----------|
| 30 | 0.0000186 | 218 | 4,298,326 | 0.0253 | 258 |
| 60 | 0.0000408 | 480 | 6,229,029 | 0.0554 | 434 |
| 120 | 0.0000602 | 708 | 8,913,541 | 0.0818 | 535 |

Based on the calculations, simulations of fuel flow through a standard and modified atomizer in a Solid Works Flow Simulation environment were undertaken (Figures 3–7).

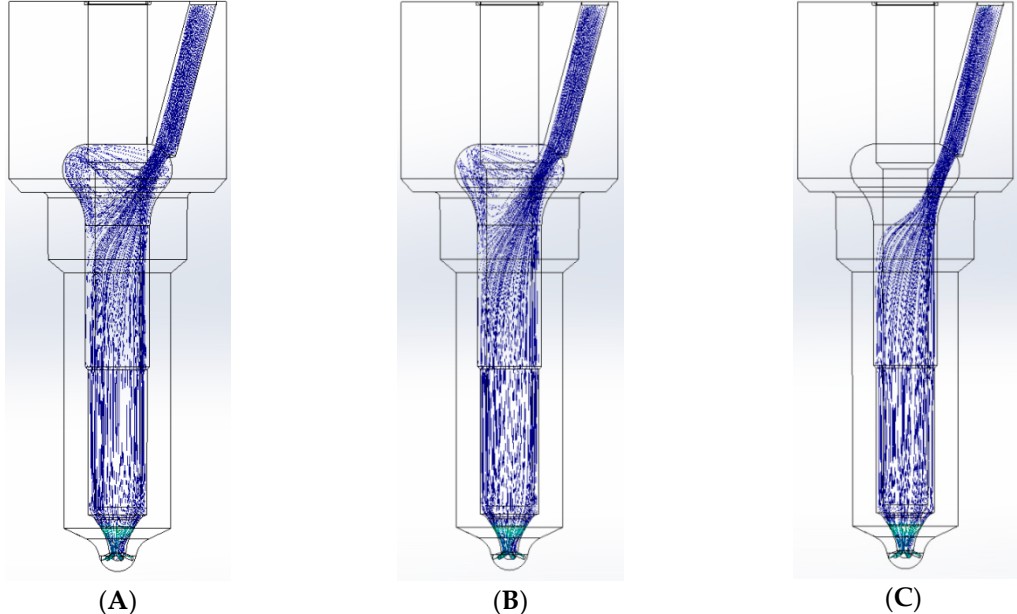

**Figure 3.** Simulation of fuel flow through a standard atomizer with an injection time of 500 μs and fuel temperature of 313 K. (**A**) System pressure of 120 MPa, injected dosage 20.9 mm$^3$/H; (**B**) system pressure of 60 MPa, injected dosage 10.2 mm$^3$/H; (**C**) system pressure of 30 MPa, injected dosage 2.8 mm$^3$/H.

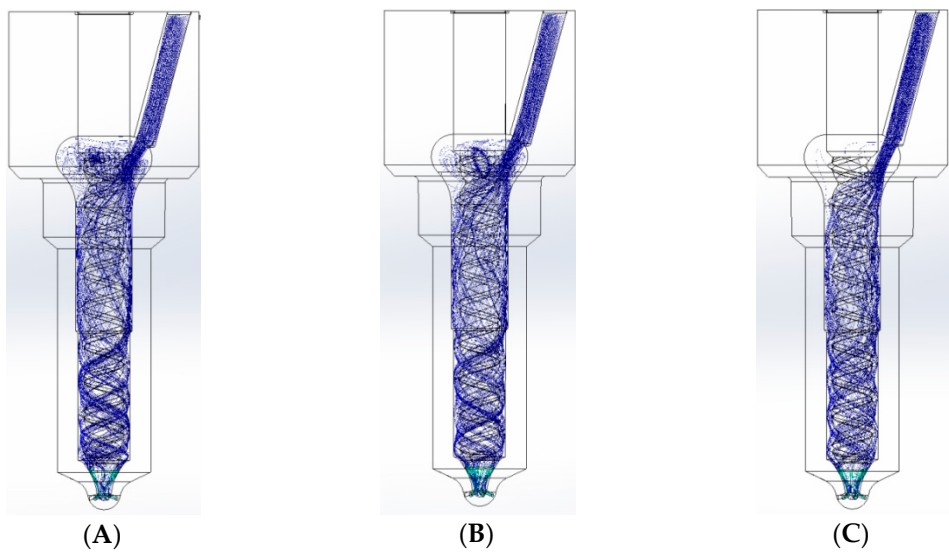

**Figure 4.** Simulation of fuel flow through a modified atomizer at 500 μs injection time and fuel temperature of 313 K. (**A**) System pressure of 120 MPa, injected dosage 20.9 mm$^3$/H; (**B**) system pressure of 60 MPa, injected dosage 10.2 mm$^3$/H; (**C**) system pressure of 30 MPa, injected dosage 2.8 mm$^3$/H.

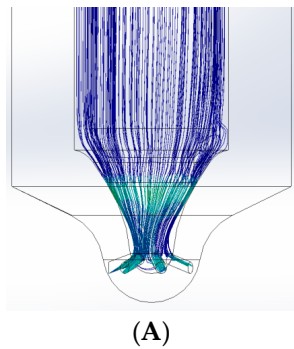

(**A**)

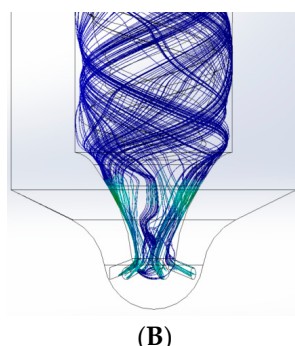

(**B**)

**Figure 5.** Simulation of fuel flow as a function of speed in the zone of valve closing and opening of the atomizer at 120 MPa system pressure. (**A**) Classical atomizer; (**B**) atomizer with spiral-elliptical ducts.

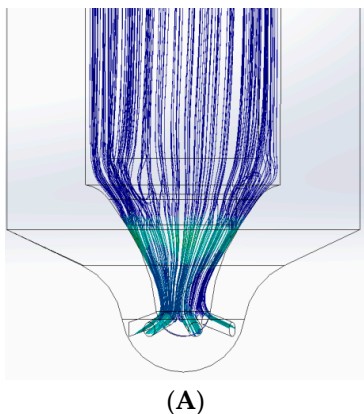

(**A**)

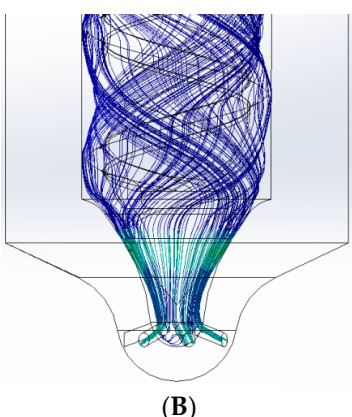

(**B**)

**Figure 6.** Simulation of fuel flow as a function of speed in the zone of valve closing and opening of the atomizer at 60 MPa system pressure. (**A**) Classical atomizer; (**B**) atomizer with spiral-elliptical ducts.

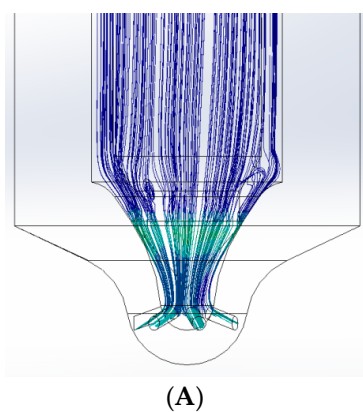

(**A**)

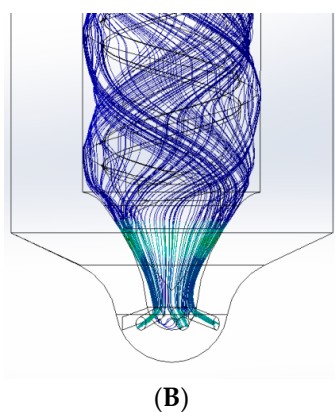

(**B**)

**Figure 7.** Simulation of fuel flow as a function of speed in the zone of valve closing and opening of the atomizer at 30 MPa system pressure. (**A**) Classical atomizer; (**B**) atomizer with spiral-elliptical ducts.

The aim of spiral-elliptical ducts is to trigger eddies along the non-working part of the injector needle. If the fuel moves in the nozzle, it will increase the temperature, due to additional turbulence. A higher temperature will lower fluid kinematic viscosity. Spiral-elliptical ducts increase the flow dimension. The Reynolds number ($R_e$) increases according to relationship (8), which has an influence on the fuel stream turbulence. The density of compressible liquids depends on fluid temperature and pressure, and can be considered within approximation (9), where $\rho_0$ is the liquid density under the

reference pressure $P_0$, C and B are coefficients, $\rho_0$, B and C depend on the temperature, and P is the calculated pressure [27].

$$\rho = \frac{\rho_0}{1 - c \cdot \ln \frac{B+P}{B+P_0}}.\tag{9}$$

Liquid dynamic viscosity depends on fluid pressure, and can be considered within approximation (10), where $\mu_0$ is the liquid density under the reference pressure $P_0$, a is a coefficient that depends on the temperature, and $P^| = 10^5$ Pa is a constant value.

$$\mu = \mu_0 e^{a(p-p|)}.\tag{10}$$

Flow simulation allows us to simulate heat transfer between solids and fluids. Heat transfer conversion has been described by energy conservation Equations (11) and (12). Heat flux is defined by Equation (13). Anisotropic solid media heat conductivity is described by Equation (14), where e is the specific internal energy, $Q_H$ is the specific heat release per unit volume, $\lambda_i$ is the eigenvalues of the thermal conductivity tensor, Pr is the Prandtl number, h is the thermal enthalpy, $\sigma_c = 0.9$ constant, u is the fluid velocity, $\Omega$ is the angular velocity, and $S_i$ is the mass distributed external force per unit mass, due to porous media resistance [27].

$$\frac{\partial \rho H}{\partial t} + \frac{\partial \rho u_i H}{\partial x_i} = \frac{\partial}{\partial x_i}(u_j(t_{ij} + t_{ij}^R) + q_i) + \frac{\partial p}{\partial t} - \tau_{ij}^R \frac{\partial u_i}{\partial x_j} + \rho\varepsilon + S_i u_i + Q_H|,\tag{11}$$

$$H = h + \frac{u^2}{2} + \frac{5}{3}k - \frac{\Omega^2 r^2}{2} - \Sigma_m h^0 y_m,\tag{12}$$

$$q_i = (\frac{\mu}{Pr} + \frac{\mu_t}{\sigma_c})\frac{\partial h}{\partial x_i}, i = 1, 2, 3,\tag{13}$$

$$\frac{\partial \rho e}{\partial t} = \frac{\partial}{\partial x_i}(\lambda_i \frac{\partial T}{\partial x_i}) + Q_H.\tag{14}$$

Figure 8 presents fuel temperature during flow through classical and spiral-elliptical duct atomizers. There is a noticeable difference in heat distribution between nozzles. The atomizer with spiral-elliptical ducts has higher fluid around the needle temperature because of additional eddies.

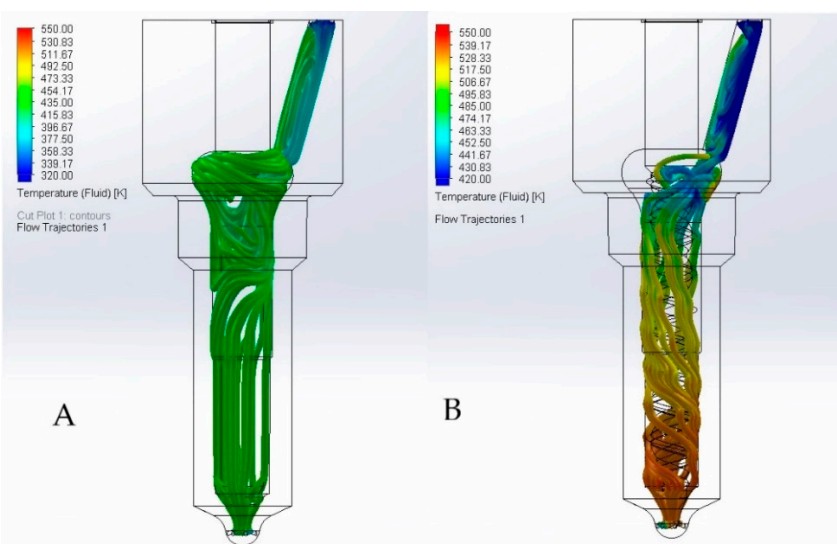

**Figure 8.** Fuel flow temperature distribution, pressure 120 MPa, fluid temperature 313 K, initial solid temperature 573 K, solid material tool steel X40Cr14. (**A**) Classical atomizer, (**B**) atomizer with spiral-elliptical ducts.

A simulation presented the fuel injector nozzle temperature distribution (Figure 9).

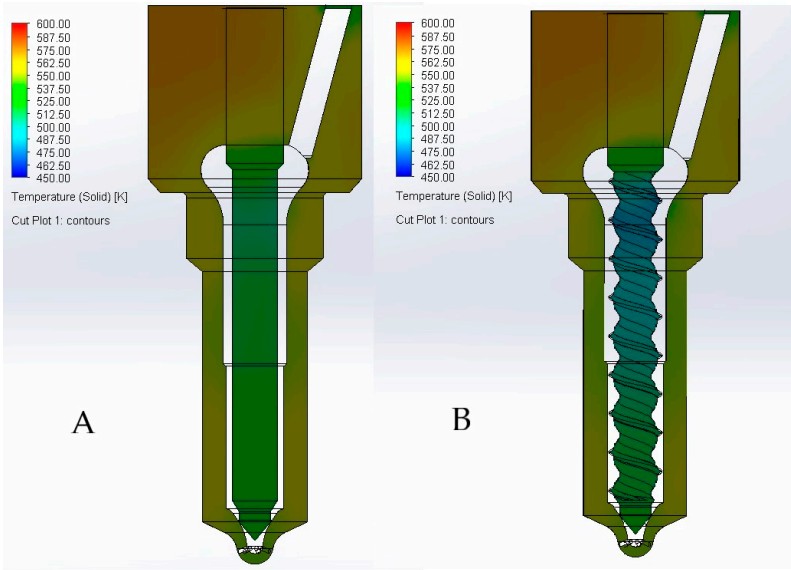

**Figure 9.** Fuel injector nozzle temperature distribution, pressure 120 MPa, fluid temperature 313 K, initial solid temperature 573 K, and solid material tool steel X40Cr14. (**A**) Classical atomizer, (**B**) atomizer with spiral-elliptical ducts.

Both nozzles have a similar temperature. The difference in the fluid temperature (Figure 8) triggered turbulence.

The research was undertaken using an infrared camera. Both injectors were working on the test bench. It was possible to compare the temperature of the chassis of standard and modified fuel injectors (Figure 10).

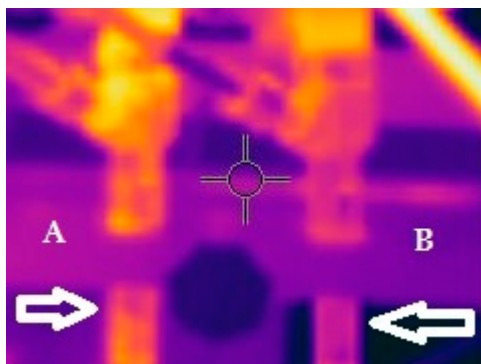

**Figure 10.** Fuel injector during test temperature distribution. (**A**) Atomizer with spiral-elliptical ducts; (**B**) classical atomizer.

The initial fuel spray visualization is shown in Figure 11. There was 30 MPa system pressure and a 300 µs injection time test. These are pilot dosage parameters for engine idle speed. This is also initial research, and the aim was to analyze if injector nozzle modification influences the fuel injected stream.

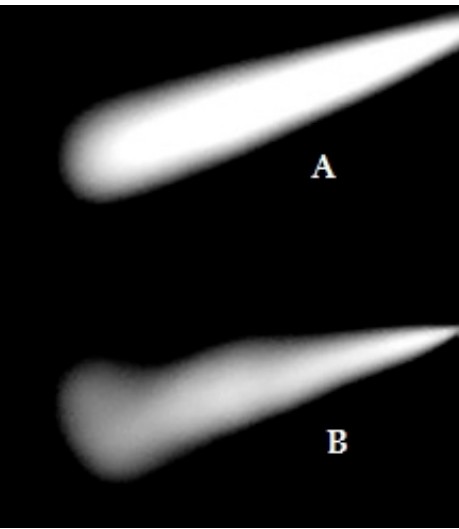

**Figure 11.** Initial fuel spray research, pressure 30 MPa, injection time 300 μs. (**A**) Fuel injector with a standard nozzle; (**B**) fuel injector with spiral-elliptical ducts.

## 5. Conclusions

Analyzing the results of simulation tests, it can be seen that modification of the non-working part of the atomizer needle changes the characteristics of the fuel flow. Spiral-elliptical ducts created additional turbulence around the spire (Figure 4). The qualitative parameters of the fuel stream depend on the velocity of the liquid at the outlet and the degree of its turbulence. The flow velocity depends on the pressure, while the degree of stream penetration is influenced by the physical parameters of the fuel, such as density and viscosity. The purpose of the modification is to intensify turbulence during fuel flow through the atomizer. The most important property of turbulent motion is the presence of many vortices of varying sizes that transfer momentum, mass and heat from one flow point to another. Vorticity $\Omega$ is the rotation of the velocity vector V. It is possible to use the kinetic energy of the rotational motion (imparted by modification of the non-working part of the needle) in order to increase the kinetic energy of the translational motion of the stream of injected fuel into the working space of the cylinder. Additionally, rotating flow around the needle has an influence on the temperature, leading to a higher Reynolds number. This phenomenon causes local changes in the physical fuel parameters, like viscosity and density. Viscosity is a parameter that affects the characteristics of the stream of fuel injected into the combustion chamber. By reducing its value as a result of the temperature rising, it is possible to improve the qualitative characteristics of the fuel injection. This process is of particular importance for a plant fueled engine, whose physical parameters are different from those of standard diesel fuel.

Qualitative parameters of the injected fuel stream play an important role during the combustion process of fuel in compression ignition engines. The results of laboratory tests have shown that the changes made to the non-working part of the fuel atomizer needle do not affect the amount of injection and overflow doses. Based on the results of the measurements, the fuel flow parameters (Reynolds number and flow speed) in the atomizer and stream were calculated. The nature of the liquid flow is determined by the Reynolds number, which depends on the speed. According to the Bernoulli equation, the results of calculations showed that in the fuel atomizer, due to high pressures, laminar fuel flow prevails (Figure 3), while in the stream the flow is hyper turbulent. The modification of the inoperative part of the needle caused additional fuel turbulence in the atomizer (Figure 4). Increased turbulence causes the temperature of the entire system to rise (Figure 8). As a result of this temperature increase, the viscosity of the fuel decreases, which affects the qualitative course of the fuel injection process. In addition, additional vortices affect the kinetic energy of the forward movement of the stream. The results of research using an infrared camera show that the temperature of the modified fuel injector is

higher than that in a standard fuel injector during tests (Figure 10). Initial fuel stream visualization (Figure 11) indicates the changes in the injected fuel quality. The fuel stream was dispersed during the injection process. The analytical simulation shows that fuel in spiral-elliptical ducts has an increased temperature, due to additional turbulence. This causes a lowering of the fuel density and dynamic viscosity. These parameters have an influence on the fuel stream shape.

Based on the theoretical analysis, it can be concluded that it is possible to change the quality parameters of the stream of injected fuel into the cylinder working space. The spiral-elliptical ducts created caused additional turbulence in the fuel flow. The proposed solution is innovative, and should find particular application in compression ignition engines fueled with vegetable fuels, due to their physical properties.

In order to further, and more comprehensively, examine the effect of spiral-elliptical ducts on the operating parameters of a compression ignition engine fueled with either conventional fuel or fuel of plant origin, the following stages of testing are planned:

1. Tests of the stream of injected fuel using an injector with a standard and modified atomizer using conventional and vegetable fuels. During laboratory tests, the quality parameters of the injected fuel stream will be analyzed.
2. Tests on the engine test bench using standard and modified fuel injectors using conventional and vegetable fuels. During the tests, the operating and ecological parameters of the compression ignition engine will be analyzed.

**Author Contributions:** Supervision, writing—review, J.E. Conceptualization, data acquisition and interpretation, investigation and writing—original draft preparation and editing, T.O.; conceptualization, data interpretation and writing—review and editing, K.F.A.; software, investigation, writing—review and editing, L.M.

**Funding:** This research received no external funding.

**Conflicts of Interest:** We do not have any personal conflicts of interest in communicating the findings of this study, and we had no sponsor who would make claims to the findings being presented.

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
