# Peer review of "Model Issues Regarding Modification of Fuel Injector Components to Improve the Injection Parameters of a Modern Compression Ignition Engine Powered by Biofuel"

_applsci, doi:10.3390/app9245479_

Round 1

Reviewer 1 Report

Article contains the important research results. The contents and results are comprehensive. In general, I support the work and believe that it can be recommended for publication. There are a number of points that have to be clarified: 1. The novelty/originality shall be justified by highlighting that the manuscript contains sufficient contributions to the new body of knowledge. The knowledge gap needs to be addressed in Abstract, Introduction, Results and Discussion, Conclusion sections. The article can be improved by analysis of research data and comparison with experimental and theoretical data from other authors. We can see many articles with research in this field. 2. Please, make special subsections in Introduction for description of the novelty and significance. 3. Please, expand the description of research procedures and errors, limits and explain the choice of research methods. 4. Please, expand the description of limited group of variated parameters. 5. It is known that biofuels are most often highly viscous compositions. The decay of droplets of such fuels is complicated. Therefore, the effects of micro-explosive dispersion are often used [Acta Astronautica. 2019. V. 160. P. 258–269]. It is advisable for the authors to expand the description of these processes and comment on whether it is possible to combine their atomization with a nozzle and micro-explosive dispersion. 6. It is advisable to expand the description of the advantages and disadvantages of nozzle devices for high viscosity fuels [Fuel. 2019. V. 254. 115606]. Correction of the manuscript will enable to improve the quality of the submitted material.

Author Response

Introduction has been changed. We described analysis of literature and significance of modifications.

Fuel injectors temperature measurements using infrared camera show rightness poses thesis that spiral – elliptical ducts in the nozzle caused fuel temperature increase inside. It confirms simulation on figure 8.

We have put under simulations figure boundary parameters like fuel temperature, injection dosage, system pressure and injection time.

We have analyzed presented articles, there are very important information. We have taken into consideration them all.

I enclose corrected article.

Reviewer 2 Report

The publication is very interesting and interesting. However, it needs to be supplemented:
The discussion should include polemics with other scientists. Change the title from discussion to summary.
The problem of fuel atomization is described in too general a way. There is no evidence of how the parameters of the fuel itself affect its flow. The authors strongly define theoretical issues and explain the essence of the problem. I believe that there is a lack of mathematical evidence to confirm the theory. The paper concerns a technical issue and should contain results of experimental research or mathematical evidence. The publications should be enriched with research results.

Author Response

We prepared temperature fuel flow simulation to show that spiral – elliptical ducts influence on turbulence inside the nozzle. I have made initial test standard and modified injector on the test bench by using infrared camera to show temperature decomposition  and spray researches for pilot dosage.

I enclose corrected article.

Reviewer 3 Report

The manuscript deals with a model analysis of a spiral-elliptical duct in a fuel injector for compression ignition engine. The authors should improve the introduction section, no references have been reported to the introduction, they can be referred to the modern fuel injection system for diesel engine, then, you should emphasize witch is the additional value of this proto-injector compared to the conventional one. There is already in production, FIS capable to work with biofuels or alternative fuels. Figure 1 is already published in some manuscript or patent? If yes, consider linking the reference. Have already the authors performed some activities on the research engine comparing the engine and emissions performance between conventional and spiral-elliptical injectors? It should be interesting to present in this manuscript the preliminary results. Overall, it is a good paper, you have designed an innovative injector that can be used as a reference in the future of CI engines. The reviewer has highlighted some points (reported above) that can improve the manuscript quality.

Author Response

I have modified introduction section. We added temperature fuel flow simulation, initial spray analysis and researches by using infrared camera. I used Figure 1 to my patent application. I added reference. I have compared engine work parameters and emission performance. We prepare second part of article and researches. Engine researches shows that carbon monoxide and soot decrease a little. These are very initial researches. We are preparing new test bench now. I would like to measure pressure  and temperature inside combustion chamber in engine working on standard and modified injectors. These results will described in second part of the paper.

I enclose corrected article.

Reviewer 4 Report

The introduction section contains no literature review. Also, the novelty of the study with respect to the state-of-the-art needs to be described better and convincingly.

Section “Impact of fuel injection parameters on the combustion process in a compression-ignition engine” can be written in a more concise way.

Line 184: expression “on the non-working part of the needle” is confusing. Also, reference is made to a “cone valve 6” apparently not shown in figure 1.

Section “Fuel flow analysis in the fuel atomizer” contains too many references. It needs to focus more on the results.

Practically no model validation was performed.

The major drawback of the work is that no spray analysis is presented.

Author Response

The introduction section has been changed. Line 184 was the type mistake. I corrected it. I have changed Fuel flow analysis in the fuel atomizer section. The literature analysis has been moved to introduction. To show the changes of temperatures in the nozzle we prepared additional model in the paper. I prepared initial spray analysis. Complex fuel spray analysis connected with engine researches we prepare in second part of researches and article.

I enclose cerrected article.

Round 2

Reviewer 1 Report

Article can be accepted.

Author Response

Thank you for an article acceptation.

Reviewer 3 Report

The authors have revised the paper according to the reviewer suggestions.

Author Response

English changes have been done by MDPI pre-editing language services. Other suggestions were considered and the paper was revised, according to the Reviewer opinion.